# Impact of Autophagy of Innate Immune Cells on Inflammatory Bowel Disease

**DOI:** 10.3390/cells8010007

**Published:** 2018-12-22

**Authors:** Tomoya Iida, Yoshihiro Yokoyama, Kohei Wagatsuma, Daisuke Hirayama, Hiroshi Nakase

**Affiliations:** Department of Gastroenterology and Hepatology, Sapporo Medical University School of Medicine, Sapporo 060-8543, Japan; tomoya.iida.0306@gmail.com (T.I.); yoshi_yokoyamaa@yahoo.co.jp (Y.Y.); waga_a05m@yahoo.co.jp (K.W.); hirarin95@yahoo.co.jp (D.H.)

**Keywords:** autophagy, innate immunity, immune cell, inflammasome, Paneth cell, inflammatory bowel disease, Crohn’s disease

## Abstract

Autophagy, an intracellular degradation mechanism, has many immunological functions and is a constitutive process necessary for maintaining cellular homeostasis and organ structure. One of the functions of autophagy is to control the innate immune response. Many studies conducted in recent years have revealed the contribution of autophagy to the innate immune response, and relationships between this process and various diseases have been reported. Inflammatory bowel disease is an intractable disorder with unknown etiology; however, immunological abnormalities in the intestines are known to be involved in the pathology of inflammatory bowel disease, as is dysfunction of autophagy. In Crohn’s disease, many associations with autophagy-related genes, such as *ATG16L1*, *IRGM*, *NOD2*, and others, have been reported. Abnormalities in the *ATG16L1* gene, in particular, have been reported to cause autophagic dysfunction, resulting in enhanced production of inflammatory cytokines by macrophages as well as abnormal function of Paneth cells, which are important in intestinal innate immunity. In this review, we provide an overview of the autophagy mechanism in innate immune cells in inflammatory bowel disease.

## 1. Introduction

The gastrointestinal tract is continuously involved in regulating the gut flora, modulating immune responses to food antigens and other substances, and maintaining homeostasis. Inflammatory bowel disease (IBD) occurs when this homeostasis is disrupted. The innate immune response is indispensable for maintaining homeostasis, and abnormal innate immune activity is deeply involved in the pathogenesis of IBD; research in this field has made substantial advancements in recent years [1,2,3]. To date, more than 200 IBD disease susceptibility loci have been identified by genome wide association studies (GWASs) [4,5]. Within these 200 loci, based upon single nucleotide polymorphism frequencies in patients with IBD versus controls, are approximately 1500 potential associated genes [6,7]. Numerous molecules involved in producing innate immune responses are also included among these loci.

Cells closely involved in the innate immune response with respect to IBD include blood cells, macrophages, dendritic cells, Paneth cells, and goblet cells, which are also involved in the pathology of IBD [8,9,10]. These cells have been found to play important roles in the development of abnormalities, including maintaining homeostasis against stress at the cellular level and modulating autophagy.

In this review, we outline the mechanism of autophagy in innate immune cells in IBD.

## 2. Pathology and Pathogenesis of IBD

IBD is a chronic inflammatory disease involving idiopathic inflammation, primarily in the gastrointestinal tract; when defined more specifically, this condition encompasses ulcerative colitis (UC) and Crohn’s disease (CD). Both are characterized by onset at a young age and the number of affected patients has risen sharply in recent years in Europe, the United States of America, and Japan [11]. Genetic predisposition (innate and acquired immunity, cytokine, and racial difference) and environmental factors (meal, drug, smoking, and infection) are greatly involved in the onset of IBD, and intestinal immune abnormalities are caused by the involvement of the state of dysbiosis, which is believed to cause IBD [11,12,13,14,15] (Figure 1).

Abnormalities related to genetic predisposition and autophagy are deeply involved in dysbiosis. Representative autophagy-related genes include *nucleotide-binding oligomerization domain containing 2 (NOD2), autophagy-related 16 like 1 (ATG16L1), and immunity-related GTPase family M (IRGM)* [16,17,18]. Autophagy has been linked to a variety of diseases; however, its link to IBD is currently the subject of much debate.

## 3. Autophagy

Autophagy is a term derived from a Greek word meaning “self-eating” and is a process that together with the ubiquitin-proteasome system, governs the degradation of intracellular proteins. In addition to immunological functions, such as antigen presentation and protection against infection, autophagy is also involved in the starvation response, carcinogenesis, and quality control of intracellular proteins and is a constitutive process necessary for maintaining proper cell homeostasis and organ health [19,20,21]. In addition to IBD, autophagy has been shown to be associated with other diseases, such as asthma [22,23,24,25], systemic lupus erythematosus [26,27], and Parkinson’s disease [28,29].

During the autophagy process, the endoplasmic reticulum or other membranous cellular structures respond to stimuli by generating a double-membrane structure called a phagophore. The ATG16L1/ATG5/ATG12 complex multimerizes and then lipidates light chain 3 (LC3)-II on this phagophore. Concurrently, the phagophore elongates to envelop the cytoplasm or organelle to be degraded, forming an autophagosome, which is a unique double-membrane organelle. The outer membrane of the autophagosome then integrates with a lysosome and forms an autolysosome. Finally, the inner membrane degrades and absorbs its contents [30] (Figure 2).

## 4. Role of Autophagy in Innate Immunity

One of the functions of autophagy is control of the innate immune response. Many studies have revealed the involvement of autophagy in innate immune reactions, and extremely precise control mechanisms and pathophysiological roles are becoming more clearly understood and have begun to be elucidated [31,32].

### 4.1. Xenophagy, Mitophagy

Innate immunity is a mechanism through which almost all multicellular organisms protect themselves from pathogens. This pathway is activated when the constructive patterns of pathogen’s components are recognized (i.e., the cell wall components of a bacterial cell or the genome of a virus). Autophagy was initially thought to be a nonspecific mechanism for degrading substances by incorporating them into a membrane structure; however, recent studies have shown that autophagosomes selectively isolate a variety of substrates through sequestosome 1-like receptors, as is observed in autophagy of pathogens (xenophagy) [33,34,35]. Although the ubiquitin-proteasome system is a well-known selective intracellular degradation system, autophagy can selectively engulf and decompose small substances, such as mitochondria, which are larger than the targets of the ubiquitin-proteasome system, indicating characteristics similar to that of mitophagy [36,37]. The major difference between autophagosomes and other membranous organelles is that autophagosomes have a dynamic structure in which necessary fractions are newly created and disappear with the digestion of contents by fusion with lysosomes; as the necessity increases, as in the starvation state, its production efficiency dramatically increases. These features are convenient for quickly carrying out quantitative control, and even when functioning to control the immune response, autophagy is more suitable than degradation by the proteasome system, and it is believed to be essential for the resolution of quantitative problems. However, when autophagy works in connection with innate immunity, the substrates to be decomposed are rarely clear except in the cases of xenophagy and mitophagy.

### 4.2. The Role of Autophagy in Inflammasomal and Type I Interferon Response

A controllable receptor tripartite motif (TRIM) protein that facilitates autophagy by recruiting autophagy-regulating factors and recognizing the target of autophagy has recently been reported as a receptor for autophagy in a new process called precision autophagy [38]. Inflammasomal and type I interferon (IFN) responses are representative components of the precision autophagic processes involved in innate immunity.

The Nod-like receptor (NLR) family of proteins, including NLRP1, NLRP3, and NLRC4, together with the apoptosis-associated speck-like protein containing a caspase recruitment domain and the protease caspase-1, functions as downstream innate inflammasomes [39] that are activated in phagocytic cells, such as macrophages, and are induced by caspase-1 via maturation of the inflammatory cytokines interleukin (IL)-1β and IL-18 and their subsequent production [40]. NLRP3 inflammasomes have attracted much attention in recent years with respect to various diseases [41,42,43,44], and numerous relationships between the state and severity of *NLRP3* gene polymorphism and IBD presentation have also been reported [45,46,47,48,49,50].

The type I IFN response plays an essential role in the innate immune response to viral infection, and RNA derived from RNA viruses invading cells can be recognized by the helicase retinoic acid inducible gene 1/melanoma differentiation-associated protein 5 and type I IFN [51]. Because signaling from type I IFN follows the Janus kinase/signal transducer and activator of transcription pathway, this cytokine has also been reported to be associated with IBD [52].

#### 4.2.1. Modulation of NLRP3 Inflammasome Suppression via Autophagy

Gram-negative bacterial lipopolysaccharides (LPSs) stimulate toll-like receptor (TLR) 4 and induce the activation of NLRP3 inflammasomes in a TIR-domain-containing adapter-inducing IFN-β (TRIF)-dependent manner with the information transfer factor. Active oxygen species derived from mitochondria are involved in the activation of NLRP3 inflammasomes. In the TRIF downstream pathway, evolutionarily conserved signaling intermediate in the Toll pathway has been reported as a factor that induces the production of reactive oxygen species (ROS) from mitochondria. This pathway is only slightly activated in wild-type macrophages. In addition, when phagolysosomes are damaged during gram-negative bacterial infection and LPS leaks into cells, NLRP3 inflammasomes are activated via a noncanonical pathway. Activation of the noncanonical pathway is thought to be induced by an unknown LPS sensor present in the cell. In addition, metabolites, such as urate crystals, cholesterol crystals, and free fatty acids, cause damage to phagolysosomes when taken up by macrophages, causing further mitochondrial damage. Adenosine triphosphate, which is known to activate surrounding cells after being released from dead cells, induces mitochondrial damage via the Purinergic 2X7 receptor (P2X7R). Influenza viral infection can cause damage to organelles, such as mitochondria and the Golgi apparatus. When damaged mitochondria produce ROS, NLRP3 inflammasomes become activated. By suppressing this series of pathways, autophagy suppresses the activation of excessive inflammasomes [53,54,55,56] (Figure 3).

Autophagy is also believed to cause the decomposition of NLRP3 and AIM2, which are constituent factors of inflammasomes [57,58], and pro-IL-1β has been reported to be degraded by autophagy [59]. Furthermore, as described above, precision autophagy via TRIM protein is also involved in the decomposition of this inflammasome constituent factor by autophagy. TRIM20 binds to NLRP3 and pro-caspase 1, recruits autophagic regulatory factors (e.g., ULK1 and Beclin1), and is degraded by autophagy [38,60,61].

Conversely, autophagy also positively regulates NLRP3 inflammasomes and is involved in both decomposition and secretion. Autophagy is involved in the secretion of cytokines, such as IL-1β, IL-18, and high mobility group box 1 protein, and when inflammasomes become activated [62], folding of IL-1β by heat shock protein 90 is necessary; the IL-1β is then secreted after being transported to the lumen of LC3-II-positive autophagy-related structures [63].

Thus, autophagy plays an important role in the innate immune response by controlling NLRP3 inflammasomes. both positively and negatively.

#### 4.2.2. Modulation of Type I IFN Responses via Autophagy

The TRIM protein regulates type I IFN responses via precision autophagy. TRIM21 binds to the IRF3 dimer, recruits autophagic regulators (e.g., ULK1 and Beclin1), and suppresses type I IFN responses by degrading this dimer via autophagy [61]. Moreover, autophagy also positively controls type I IFN responses. Intracellular viral replication intermediates on endosomes are recognized by TLR7 and cause a type I IFN response, but autophagy promotes type I IFN responses by this TLR7 via viral recognition [64]. As such, autophagy also positively and negatively controls type I IFN responses.

## 5. Role of Autophagy of Innate Immune Cells in IBD

As discussed above, autophagic abnormalities are deeply involved in the pathology of IBD, particularly CD. Recently, GWASs identified several genetic variants, including variants of *NOD2* [53,65,66,67,68], *ATG16L1* [69,70,71,72,73,74,75,76,77], *IRGM* [78,79,80,81], and *XIAP* [82,83,84,85,86,87] linked to the onset of CD. *NOD2* is the first disease-susceptibility gene discovered for CD [16]. Abnormalities in the *NOD2* gene are found primarily in Westerners and not in Asians. NOD2 is an intracytoplasmic pattern recognition receptor belonging to the NLR family, which recognizes and defends against pathogens and foreign components invading the cytoplasm [88]. NOD2 is strongly expressed by macrophages and dendritic cells (DCs), particularly Paneth cells, and functions through mechanisms involving autophagy, intracellular bacterial sensing, modulation of the antibacterial peptide α-defensin in the Paneth cells of the small intestine, and improvement of immune tolerance by suppressing TLR signals [89]. ATG16L1 is a homolog of ATG16, the existence of which was first reported by Mizushima et al. [69,70]. The *ATG16L1* gene is involved in host immune responses against intracellular parasitic bacteria and viruses via autophagy. In particular, the *ATG16L1* gene is closely associated with Paneth cells and plays an important role in CD pathology. In addition, many genes are associated with various pathologies of IBD in connection with autophagy, and these genes play important roles in immune cells, such as macrophages and DCs, as well as intestinal epithelial cells, including Paneth cells. The relationship between immune cells and autophagy in the IBD state is described below.

### 5.1. Hematopoietic Cells

#### 5.1.1. Macrophages and DCs

Macrophages and DCs phagocytose foreign substances and bacteria in different tissues, serve as a first line of defense, and act as antigen-presenting cells to exert the functions of the acquired immune system [90,91]. Given the key role of the interactions between host and microbes in the intestine, it is critical to properly regulate pattern recognition receptor (PRR) signals and cytokine secretion. The NLRs in the cytoplasm and TLRs on the cell surface are the two main types of PRRs in innate immune cells [92]. NLRs and TLRs in macrophages are closely associated with autophagy, and macrophage autophagy is highly related to the mediation of innate immune responses in the intestinal wall [16,17]. Additionally, various antigens are degraded by the actions of proteasomes and lysosomes and are then presented by macrophages and DCs via class I and class II MHCs, after which the adaptive immune system is activated. Although it remains unclear as to how intracellular antigens are delivered to lysosomes and decomposed, recent studies have shown that autophagy is deeply involved in this process [93]. In the following section, we describe the relationships among macrophages, DCs, and autophagy in the pathology of IBD with respect to three components: pathogen degradation, suppression of inflammatory cytokine secretion, and antigen presentation.

##### Pathogen Degradation

NOD2 recruits the autophagy protein ATG16L1 to the plasma membrane at the bacterial entry site; mutant NOD2 fails to recruit ATG16L1 to the plasma membrane, and the wrapping of invading bacteria by the autophagosome is impaired. Thus, patients with CD with *NOD2* variants exhibit autophagy-related disorders [65,66,67,68]. Additionally, the capacity for autophagy and the phagocytosis of pathogenic bacteria become impaired in macrophages harboring mutant *NOD2* [94]. Furthermore, autophagy has been reported to be impaired in macrophages transfected with siRNA targeted to ATG16L1 or IRGM, and intracellular adherent-invasive *Escherichia coli* (AIEC) populations increase in the presence of intraperitoneal macrophages in *NOD2*-deficient mice [95]. Moreover, death-receptor activation or starvation-induced metabolic stress in human and murine macrophages increases the degradation of T300A or T316A variants of *ATG16L1*, respectively, resulting in diminished autophagy. In addition, knock-in mice harboring the *ATG16L1* T316A variant show defective clearance of the ileal pathogen *Yersinia enterocolitica* [74]. In an experiment using a cell line originating from macrophages, infection with CD-associated AIEC or administration of LPS or muramyl dipeptide induces IRGM expression [81]. Macrophages and epithelial cells lacking *GPR65* exhibit impaired autophagy for clearance of intracellular bacteria [96].

##### Suppressing Inflammatory Cytokine Secretion

*ATG16L1*-deficient macrophages are known to overproduce the inflammatory cytokine IL-1β in response to LPS stimulation, and the importance of ATG16L1 in inflammation has been described based on the finding that intestinal inflammation, induced by dextran sodium sulfate (DSS) intake in bone marrow chimeric mice, results in IL-1β overproduction by hematopoietic cells, leading to increased inflammation and cell fragility [53]. Moreover, in macrophages and DCs, autophagy controls IL-1β secretion by mediating the degradation of pro-IL-1β [59]. Furthermore, macrophages lacking ATG7, another autophagy-related factor, and macrophages that inhibit the activity of Vps34, which is essential for the induction of autophagy, also produce excessive quantities of IL-1β in response to LPS. Macrophages lacking ATG5 have also been reported to produce IL-1β in excess in response to the presence of gram-negative bacteria [97]. These phenomena are caused by the activation of NLRP3 inflammasomes due to autophagic disorders, and in recent years, activation of NLRP3 inflammasomes in response to various stimuli has been found to be controlled by autophagy [54,55,56] (Figure 3). In another recent report, impairing autophagy using ATG5 siRNA or an autophagy inhibitor (3-MA) was found to induce more robust initiation and activation of the NLRP3 inflammasome combined with increased caspase-1 activation and IL-1β production in peritoneal macrophages treated with LPS/DSS. 3-MA has also been shown to aggravate symptoms of DSS-induced colitis [98].

Loss of the autophagy-related gene *ATG16L1* has been shown to promote accumulation of the adaptor TRIF and enhance production of IFN-β and IL-1β as downstream signaling molecules in macrophages [99]. Macrophages from IBD risk carriers show increased myotubularin-related protein 3 expression and, in turn, decreased autophagy and increased cytokine secretion [100]. DSS and *Saccharomyces cerevisiae* were also inoculated into mice deficient in *ATG16L1*, an autophagy-related gene specific to CD11c+ DCs. Colitis has been reported to be exacerbated by elevated levels of IL-1β and tumor necrosis factor (TNF)-α when exposed to *Salmonella typhimurium* [101].

##### Antigen Presentation

In DCs, autophagy controls how antigens are processed and presented for antigen presentation [93]. Both *NOD2* 1007fs and *ATG16L1* T300A block muramyl dipeptide induction of autophagy, and this process is associated with defective bacterial handling in DCs and impaired antigen presentation in association with MHC class II at the cell surface [65]. Deletion of *ATG16L1* in a mouse model resulted in increased T-cell stimulation by DCs [102]. Additionally, *ATG7*-deficient mouse DCs are unable to stimulate CD4+ T-cell activation when exposed to *Toxoplasma gondii* antigens [103].

#### 5.1.2. Neutrophils

Neutrophils have strong phagocytic and bacterial killing ability against foreign materials and bacteria, similar to macrophages, and play a central role in innate immunity [104]. Neutrophils react sensitively to stimuli and exhibit various functions. The initial response of neutrophils is quick, and neutrophils are subjected to sophisticated control mechanisms because they are involved in tissue restoration and minimization of injury to surrounding tissues. When this balance is impaired, neutrophils exhibit abnormal activation and are involved in various diseases [105]. Neutrophils are also involved in IBD pathophysiology and have been reported to be associated with cytokines, chemokines, ROS, and elastase [106]. Several reports have described the relationship between neutrophils and autophagy in IBD. For example, impaired *Salmonella typhimurium* clearance and increased ROS production were observed in *ATG16L1*-deficient murine neutrophils [107]. Thus, autophagy-related *ATG16L1* is essential for bacterial clearance and suppression of ROS production by neutrophils. In addition, autophagy receptor optineurin-deficient mice have been shown to be more susceptible to *Citrobacter* colitis and *E. coli* peritonitis and showed reduced levels of TNF-α in serum and diminished neutrophil recruitment to sites of acute inflammation compared with that in wild-type mice [108]. Thus, the autophagy receptor optineurin plays a role in acute inflammation and neutrophil recruitment.

#### 5.1.3. Innate Lymphoid Cells (ILCs)

ILCs are innate immunocompetent cells belonging to the lymphocyte system; some are similar to helper T cells, although cytotoxic natural killer cells (NKs) are also considered ILCs. ILCs are classified into groups 1–3 based on cytokine production. Group I ILCs include ILC1s and NK cells, ILC2s are assigned to group 2 ILCs, and group 3 ILCs include ILC3s. These groups of cells have attracted much attention from researchers interested in IBD pathology and are important for the maintenance of homeostasis and inflammatory immune responses [109]. In addition, autophagy has been shown to be required for ILC development and function. Atg5, an essential component of the autophagy machinery, is required for the development of mature NKs and group 1–3 ILCs [110]. Phosphorylated Forkhead box O (FoxO) 1 is localized in the cytoplasm of immature NKs and interacts with ATG7. FoxO1-mediated autophagy has been shown to be required for NK development and NK-induced innate immunity [111,112].

#### 5.1.4. NKT Cells (NKTs)

NKTs are immunocompetent cells that recognize antigens presented mainly on CDld molecules of antigen-presenting cells and produce cytokines. Because CD1d-restricted NKTs have an invariant T-cell receptor α chain, these cells are expressed as intramucosal NKTs (iNKTs). Mice with oxazolone-induced enteritis are considered a Th2-dominant ulcerative colitis model, and IL-13 produced from iNKTs is an associated cytokine [113]. It has been reported that the number of NKTs is increased in the intestinal mucosa of patients with ulcerative colitis, and IL-13 production is enhanced [114]. Thus, NKTs are thought to be involved in the pathogenesis of IBD, and autophagy is important for the development and differentiation of iNKTs [115,116]. Recent reports have demonstrated that IL-15 induces autophagy of NKTs via TBK-binding protein 1 [117].

### 5.2. Intestinal Epithelial Cells

#### 5.2.1. Paneth Cells

Paneth cells are found at the base of small intestinal crypts and were first reported in 1888 as epithelial cells with dense coarse granules. Subsequently, α-defensin, an antimicrobial peptide and an innate immune effector, was discovered within Paneth cell granules [118]. Paneth cells play an important role as intestinal epithelial cells responsible for innate immunity [119,120,121]. In Caucasians, abnormalities in the *NOD2* gene, which was first identified as a CD susceptibility gene, inhibit α-defensin secretion by Paneth cells [122,123]. In addition, in mice with low ATG16L1 expression, significant abnormalities were observed in the secretory pathway of Paneth cell granules, and in patients with CD harboring homozygous *ATG16L1* T300A mutations, abnormal Paneth cells, similar to low ATG16L1 expression mice, were reported to have been found in noninflammatory sites in the ileum [72]. In addition, as described above, autophagic abnormalities are involved in the pathology of CD, and abnormal control of endoplasmic reticulum stress (ERS), ROS, and gut flora by Paneth cells each play important roles.

##### ERS Stress (ERS)

Various intracellular proteins are synthesized in the ER, and these proteins undergo proper folding and are transported to the Golgi apparatus. Unfolded or misfolded proteins accumulate in the ER. The accumulation of proteins with these conformational abnormalities is called ERS, and excessive ERS ultimately induces cell apoptosis. The homeostatic mechanism mediating excessive ERS is known as the unfolded protein response (UPR). The UPR plays an important role in the survival and function of intestinal epithelial cells. In recent studies, the UPR has been shown to control abnormalities involved in the pathogenesis of IBD, and several genes associated with ERS have been reported as disease susceptibility genes by GWASs [12,13,21,89].

As intestinal epithelial cells, Paneth cells are closely related to ERS [12]. ERS results in the induction of autophagy in Paneth cells through three signaling pathways: insulin response element (IRE) 1/c-Jun N-terminal kinase/nuclear factor-κB/X-box binding protein 1 (XBP-1), pancreatic ER kinase/eukaryotic initiation factor (eIF) 2α-activated transcription factor 4, and GRP78-activated transcription factor 6/CCAAT-enhancer-binding protein homologous protein signaling pathways [124,125,126]. In terms of genetic abnormalities, reports describing the relationships between ERS and autophagy in Paneth cells have focused on the roles of the *NOD2* and *ATG16L1* genes [12,122,127,128,129]. Secretory autophagy, which is triggered in Paneth cells by bacteria-induced ERS and limited bacterial dissemination, is disrupted in Paneth cells expressing *ATG16L1* T300A [122]. In mice lacking *ATG16L1* specifically in the epithelium, CD-like ileitis occurs spontaneously in an age-dependent manner in Paneth cells, which are ERS sensors and causes changes in IRE1α, which is important for Paneth cell homeostasis [129]. ER stress is induced via deletion of the UPR transcription factor XBP-1 in the intestinal epithelium, resulting in autophagosome formation in Paneth cells via a mechanism involving the eukaryotic translation initiation factor eIF2α [8]. Thus, the involvement of autophagy in ERS may be a new therapeutic target for IBD in the future.

##### ROS

ROS have been shown to be involved in the pathogenesis of IBD, and many reports have suggested that IBD is associated with an imbalance between ROS and antioxidant activity, causing oxidative stress as a result of either ROS overproduction or decreased antioxidant activity [130,131,132]. Although many of the details of the relationship between ROS and autophagy have not been elucidated, accumulation of ROS has been reported to be related to induction of autophagy [133]. Notably, mitochondrial dysfunction is known to trigger the accumulation of ROS in Paneth cells, resulting in induction of autophagy through the p53/TP53 induced glycolysis regulatory phosphatase/damage-regulated autophagy modulator, p62-NF-E2-related factor 2, and BCL2/adenovirus E1B 19 kDa protein-interacting protein 3 pathways, thus protecting against cellular damage caused by various stresses [134,135]. Additionally, mutations in Atg promote the production of ROS via mitochondrial insufficiency in Paneth cells [136]. Further studies are needed to fully elucidate the relationship between ROS and autophagy in IBD.

##### Gut Microbiota

There are in the order of 100 trillion intestinal bacteria present in the human intestinal tract, and these cells play important roles in maintaining host metabolism and immunological homeostasis. Although findings suggesting the involvement of intestinal bacteria in the pathology of IBD are accumulating, it is unclear whether changes in the composition of the gut flora are the cause of IBD. Paneth cells are important factors influencing the gut flora [137], and autophagy abnormalities in Paneth cells are known to be related to alterations in the gut flora [137,138,139]. Autophagy dysfunction in Paneth cell disrupts the normal intestinal flora and promotes intracellular survival of AIEC and *Salmonella typhimurium* [138]. Vitamin D receptors in the intestinal tract contribute to Paneth cell function, autophagy function, and maintenance of normal intestinal microflora via ATG16L1 [139]. In addition, the microbiota induces basal Paneth cell autophagy by IFN-γ, facilitating the maintenance of intestinal homeostasis [140]. Dysfunction of autophagy by Paneth cells has also been reported to cause not only changes in the gut microbiome, but also improper responses to the altered flora [141]. As described above, the gut flora is closely related to autophagic abnormalities in Paneth cells, and many IBD treatments, particularly those utilizing probiotics, aim to correct intestinal bacterial flora populations [142,143].

#### 5.2.2. Goblet Cells

Goblet cells are intestinal epithelial cells with cytoplasmic granules containing large quantities of mucin, a glycoprotein [144,145]. In the gastrointestinal tract, Goblet cells can be found in both the small intestine and the large intestine, and mucin secreted into the intestinal lumen is thought to contribute to mucosal protection and repair of the intestinal epithelium [146]. Dysfunction in mucin secretion and defects in the mucus layer allow large quantities of bacteria to reach the epithelium and trigger excess host immune responses, which have been shown to be associated with IBD [147].

In recent studies, autophagy has been reported to affect the functions of Goblet cells. Mice or cells lacking autophagy by depletion of autophagy-related proteins, such as ATG5, ATG7, ATG16L1, and LC3, show altered goblet cell morphology and decreased mucin secretion [73,148,149]. In addition, inflammasomes have been shown to be involved in mucin secretion by Goblet cells, and NLRP6 inflammasomes have been reported to promote excocytosis of mucin by Goblet cells through promotion of autophagy [150]. Sonic hedgehog intestinal epithelial conditional knockout mice showed decreased numbers of ileal mucin-secreting Goblet cells accompanied by a significant reduction in autophagy [151]. Apple polysaccharide has been shown to inhibit dysbiosis-associated gut permeability and chronic inflammation due to the induction of autophagy in Goblet cells [152]. As described above, Goblet cells play important roles in the secretion of mucin via autophagy, act to maintain intestinal microflora, and are attracting attention as new therapeutic targets for IBD [149,153,154].

An overview of the autophagy mechanism of innate immunity cells in IBD is shown in Figure 4.

## 6. Conclusions

In this review, we have provided an overview of the autophagy mechanism of innate immune cells in IBD. Numerous studies have revealed that autophagy is an essential key player in controlling inflammatory responses mediated by innate immune cells. We anticipate that in the future, and based on further research, a therapy based on autophagic control will be established for IBD, a condition that does not yet have a curative therapy.

## Figures and Tables

**Figure 1 cells-08-00007-f001:**
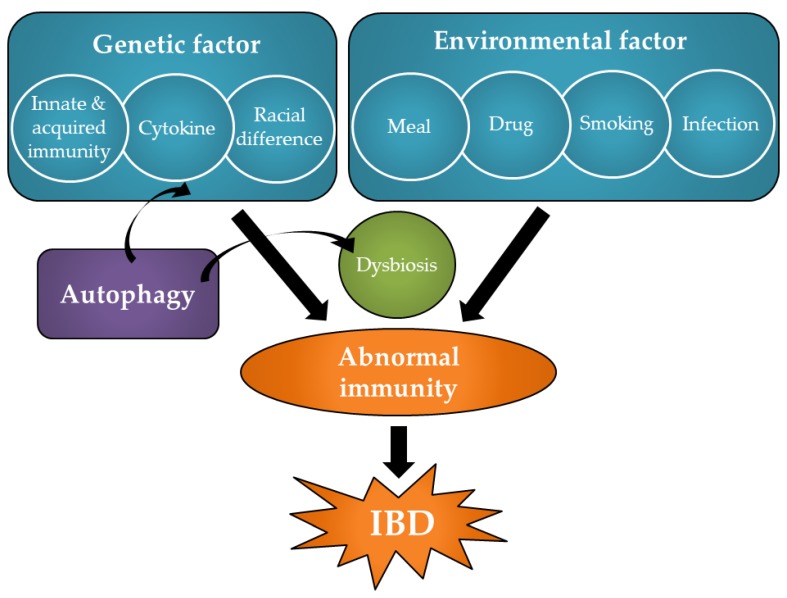
Pathology and pathogenesis of inflammatory bowel disease (IBD). Genetic predisposition and environmental factors are greatly involved in the onset of IBD, and intestinal immune abnormalities are caused by the involvement of the state of dysbiosis, which is believed to cause IBD.

**Figure 2 cells-08-00007-f002:**
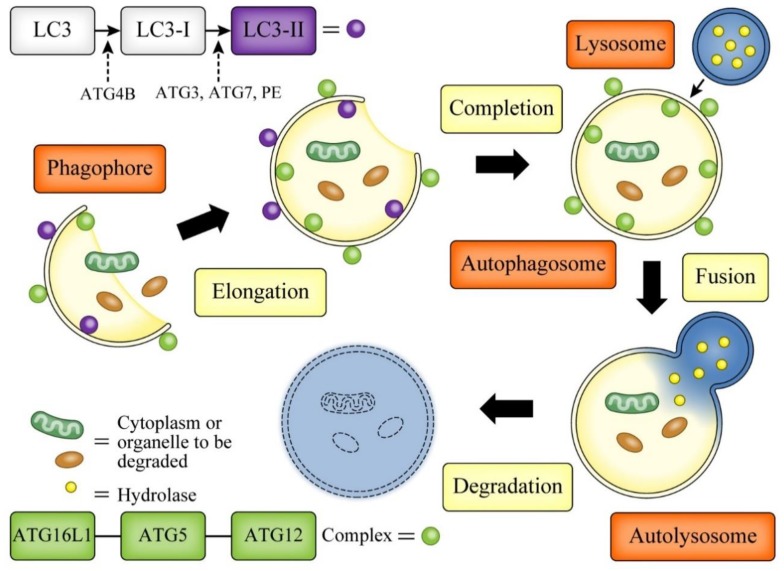
Autophagy mechanism. The endoplasmic reticulum or other membranous cellular structures respond to stimuli by generating a double-membrane structure called a phagophore. ATG16L1-ATG5-ATG12 complex multimerizes and then lipidates light chain 3 (LC3)-II on this phagophore. Concurrently, the phagophore elongates to envelop the cytoplasm or organelle to be degraded, forming an autophagosome. The outer membrane of the autophagosome then integrates with a lysosome and forms an autolysosome. Finally, the inner membrane degrades and absorbs its contents.

**Figure 3 cells-08-00007-f003:**
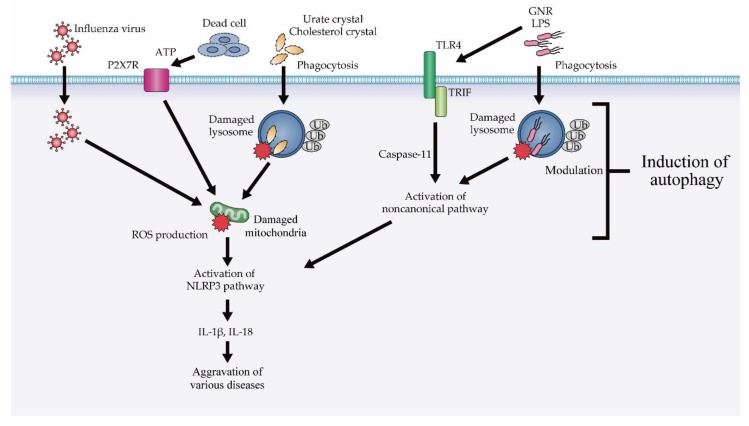
Modulation of NLRP3 inflammasome suppression via autophagy, ATP; adenosine triphosphate, GNR; gram negative rods, IL; interleukin, LPS; lipopolysaccharide, P2X7R; Purinergic 2X7 receptor, ROS; reactive oxygen species, TLR4; Toll-like receptor 4, TRIF; TIR-domain-containing adapter-inducing interferon-β, Ub; ubiquitin.

**Figure 4 cells-08-00007-f004:**
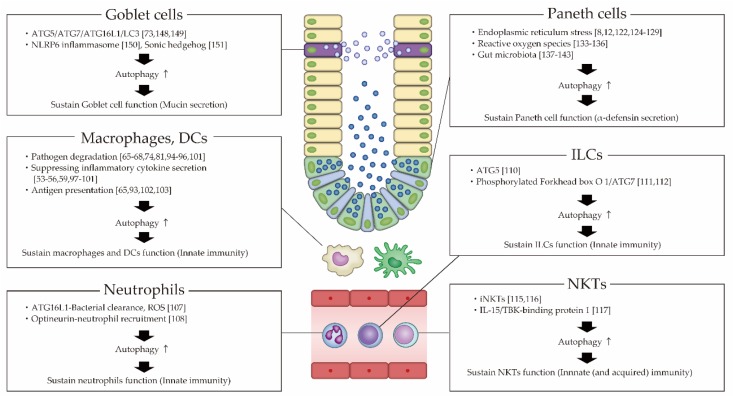
Role of autophagy of innate immune cells on IBD. DCs; dendritic cells, IL; interleukin, ILCs; innate lymphoid cells, NKTs; natural killer T cells, ROS; reactive oxygen species.

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
