# Peer review of "Impact of Autophagy of Innate Immune Cells on Inflammatory Bowel Disease"

_cells, 2018, doi:10.3390/cells8010007_

Round 1

Reviewer 1 Report

 The review by Iida et al, provides an overview on the role of autophagy of innate immune cells on inflammatory bowel disease showing as the intracellular degradation mechanism has many immunological functions. Autophagy is an essential key player in controlling inflammatory response mediated by innate immune cells.

The review is well structured. In the first chapter is described how genetic predisposition and environmental factors are involved in altering immune homeostasis thus inducing the onset of dysbiosis which is  believed to cause IBD.

Then autophagy mechanism is described and analyzed it’s role in innate immunity .

In the last chapter is described in particular the role of the innate immune cell on IBD, showing  the relationship of  the different element and autophagy in IBD 

The bibliography is well articulated and comprehensive of many references quite recent.

But it doesn’t contributes to much to  knowledge since  its design and its structure is very similar to review of reference n.153. El-Khider, F.; McDonald, C. Links of Autophagy Dysfunction to Inflammatory Bowel Disease Onset. Dig. 795 Dis. 2016, 34, 27-34. [DOI: 10.1159/000442921]

Author Response

But it doesn’t contributes to much to  knowledge since  its design and its structure is very similar to review of reference n.153. El-Khider, F.; McDonald, C. Links of Autophagy Dysfunction to Inflammatory Bowel Disease Onset. Dig. 795 Dis. 2016, 34, 27-34. [DOI: 10.1159/000442921]

Thank you for your review and your great comments. However, I don’t think that the design and the structure of this manuscript is similar to those of reference No.153. Our manuscript was reviewed from a new viewpoint that described the relationship between autophagy and neutrophils, ILCs, and NKTs. Since these contents were not described in detail in the previous reviews, we think this review is worth being newly described.

Reviewer 2 Report

This is an interesting review and perhaps one of a few published studies, which demonstrates the role of autophagy of innate immune cells on inflammatory bowel disease. The authors covered most of the parts and I would recommend to consider it for publication.

Author Response

This is an interesting review and perhaps one of a few published studies, which demonstrates the role of autophagy of innate immune cells on inflammatory bowel disease. The authors covered most of the parts and I would recommend to consider it for publication.

Thank you for your review and your great comment. We think our manuscript would be acceptable for the Cells journal.

Reviewer 3 Report

The manuscript entitled “Impact of autophagy of innate immune cells on inflammatory bowel disease” presents interesting issue, but some important corrections are needed.

Main problems:

(1)    The serious flaw of the presented manuscript is associated with the fact, that it presents a highly subjective review, not a systematic review. While the systematic review has a key role for broadening knowledge, the other reviews don’t have such role. Authors should indicate that they conducted a narrative review, in order to present specific information for readers.

(2)    Taking into account, that the Materials and methods section is not presented (it should be added), without any specific information, it is hard to understand which studies were included into review and why. Authors did not present any key words, which were used during literature search, inclusion and exclusion criteria of references, information about the procedure of literature search conducted by them, number of chosen references, as well as information if some of them were excluded from the review and on the basis of which criteria.

(3)    Including the reviews into own review is also a highly controversial procedure – in many aspects, Authors just repeated the conclusions of other authors, without own analysis or conclusions.

(4)    In the case of presented figures (Figure 1 – Figure 3) Authors should precisely indicate if they are one and only authors or if it is the modified version of figure of the other Authors (in such situation, the reference is needed)

Minor comments:

(1)    Authors in their abstract justified their study, but they did not present any information associated with the applied methodology (in the case of review articles it should be presented also) – see above

(2)    While specifying the etiology of IBD, Authors should also indicate the role of diet.

(3)    The Author Contributions Section seem to be confusing – 3 Authors (YY, KW, DH) participated only in “investigation” (what do Authors mean by “investigation” in the case of review article?) – how was it possible, that they did not participate in the original draft preparation? If they conducted literature search it means, that they participated in the original draft preparation. But if not, Authors who did not participate in the manuscript preparing should be removed and not presented as an author (due to the risk of guest authorship procedure that is forbidden) and presented in Acknowledgements section only.

Author Response

Main problems:

(1) The serious flaw of the presented manuscript is associated with the fact, that it presents a highly subjective review, not a systematic review. While the systematic review has a key role for broadening knowledge, the other reviews don’t have such role. Authors should indicate that they conducted a narrative review, in order to present specific information for readers.

Thank you for your comment. Instruction of authors of Cells shows that reviews provide concise and precise updates on the latest progress made in a given area of research. We described this manuscript objectively following this instruction.

(2) Taking into account, that the Materials and methods section is not presented (it should be added), without any specific information, it is hard to understand which studies were included into review and why. Authors did not present any key words, which were used during literature search, inclusion and exclusion criteria of references, information about the procedure of literature search conducted by them, number of chosen references, as well as information if some of them were excluded from the review and on the basis of which criteria.

Thank you for your comment. Although we read many reviews of Cells, none of them described the Materials and Methods section.

(3) Including the reviews into own review is also a highly controversial procedure – in many aspects, Authors just repeated the conclusions of other authors, without own analysis or conclusions.

Thank you for your comment. Instruction of authors of Cells shows that reviews provide concise and precise updates on the latest progress made in a given area of research. We described this manuscript objectively following this instruction.

(4) In the case of presented figures (Figure 1 – Figure 3) Authors should precisely indicate if they are one and only authors or if it is the modified version of figure of the other Authors (in such situation, the reference is needed)

Thank you for your comment, but we are sorry that we couldn’t understand your thoughts. Figure 1, 2, and 3 referred No.11-15, 30, and 53-56, respectively in the manuscript. Isn’t this enough for you?

Minor comments:

(1) Authors in their abstract justified their study, but they did not present any information associated with the applied methodology (in the case of review articles it should be presented also) – see above

(2) While specifying the etiology of IBD, Authors should also indicate the role of diet.

Thank you for your comment. Although I don’t think that diet is necessary for the main theme, we added some words in Pathology and Pathogenesis of IBD section.

(3) The Author Contributions Section seem to be confusing – 3 Authors (YY, KW, DH) participated only in “investigation” (what do Authors mean by “investigation” in the case of review article?) – how was it possible, that they did not participate in the original draft preparation? If they conducted literature search it means, that they participated in the original draft preparation. But if not, Authors who did not participate in the manuscript preparing should be removed and not presented as an author (due to the risk of guest authorship procedure that is forbidden) and presented in Acknowledgements section only.

Thank you for your comment. According to your comment, we corrected this section.